# The Identification of Seven Chemical Warfare Mimics Using a Colorimetric Array

**DOI:** 10.3390/s18124291

**Published:** 2018-12-06

**Authors:** Michael J. Kangas, Adreanna Ernest, Rachel Lukowicz, Andres V. Mora, Anais Quossi, Marco Perez, Nathan Kyes, Andrea E. Holmes

**Affiliations:** Department of Chemistry, Doane University, Crete, NE 68333, USA; adreanna.ernest@doane.edu (A.E.); rachel.lukowicz@doane.edu (R.L.); andres.mora@doane.edu (A.V.M.); anais.quossi@gmail.com (A.Q.); marco.perez@doane.edu (M.P.); nathan.kyes@doane.edu (N.K.)

**Keywords:** warfare mimics, mustard gas, colorimetric array, principle component analysis, RGB data

## Abstract

Chemical warfare agents pose significant threats in the 21st century, especially for armed forces. A colorimetric detection array was developed to identify warfare mimics, including mustard gas and nerve agents. In total, 188 sensors were screened to determine the best sensor performance, in order to identify warfare mimics 2-chloro ethyl ethylsulfide, 2-2′-thiodiethanol, trifluoroacetic acid, methylphosphonic acid, dimethylphosphite, diethylcyanophosphonate, and diethyl (methylthiomethyl)phosphonate. The highest loadings in the principle component analysis (PCA) plots were used to identify the sensors that were most effective in analyzing the RGB data to classify the warfare mimics. The dataset was reduced to only twelve sensors, and PCA results gave comparable results as the large data did, demonstrating that only twelve sensors are needed to classify the warfare mimics.

## 1. Introduction

Improving the non-presumptive detection of chemical warfare agents for soldiers in the field is a pressing need [1]. Warfare agents include chemical and explosive compounds, such as nerve agents like sarin, soman, and tabun, which are clear, colorless liquids without strong odors, and can cause loss of consciousness, seizures, and eventual death [2]. Other chemicals of war include vesicants or poisons, such as mustard gas, lewisite, phosgene, phosgene oxime, cyanide, mace, ricin, and pepper spray made of capsaicin [2]. Additionally, the detection of the precursors and degradation products of these materials can be equally important [3]. Existing technology has shown that these devices can effectively detect such analytes, but requires high-tech, expensive, and large instrumentation, which makes them less than ideal for field applications [4,5,6,7].

Colorimetric and fluorescent sensors have the potential to be effective detection systems for these analytes, but require minimal instrumentation. New sensors for these analytes have recently been reviewed [8]. One drawback of sensors is that they can be limited in the versatility or specificity of the analytes that can be identified. This can be overcome by combining multiple sensors into a sensor array, and the pattern of color changes can be used to identify and quantify a wide range of analytes [9,10,11,12]. A few sensor arrays chromofluorogenic supramolecular complexes for the chromogenic or fluorogenic sensing of nerve agents have been reported [13], as well as color changes of organophosphorus warfare mimics in the gas phase [14,15].

Exposure guidelines by the Center Disease Control suggest that mustard gas concentrations of 3.9 mg/m^3^ cause life-threatening effects or death. Therefore, there is a pressing need to identify sensors that can quickly detect this life-threatening agent. Herein, we examined the screening of 188 colorimetric sensors, in solution, to simulate an array for the detection of 144 samples of mustard gas mimics, 2-chloro-ethyl-ethylsulfide (0.1 M), 2-2′-thiodiethanol (1 M), and nerve agent mimics, trifluoroacetic acid (1 M), methylphosphonic acid (1 M), dimethylphosphite (1 M), diethylcyanophosphonate (1 M), and diethyl (methylthiomethyl)phosphonate (0.1 M) (Figure 1) [14,15,16]. The identity and CAS numbers of all 188 sensors are located in the Appendix A.

## 2. Materials and Methods

All reagents were purchased from various chemical supply companies at technical grade or better, and were used as received. The universal indicator was composed of Van Urk’s and Yamada’s recipe, and contained methyl red, methyl orange, phenolphthalein, and bromothymol blue powders in a weight ratio of approximately 1:3:7:8 [17]. One percent weight-by-weight solutions of all sensors were prepared by dissolving each into aliquots of a solvent mixture consisting of acetate buffer (0.1 M, pH 5), ethylene glycol, triethylene glycol monobutyl ether, and glycerol, in a ratio of 14:1.6:1:3.2. The sensors were dissolved by sonication in a bath sonicator (30 °C) for 1 h, followed by 5 min mixing with a probe sonicator, and then vacuum filtered once through Whatman #1 filter paper, and once through Whatman nylon filter membranes (0.2 µm).

A solution of methyl phosphonic acid (1 M) was prepared dissolving an appropriate amount of the analytes in milli-Q water (18 MΩ-cm). Solutions of 2 chloroethyl ethyl sulfide (0.1 M), 2,2′ thiodiethanol (1 M), diethylcyanophosphonate (0.1 M), dimethylphosphite (1 M), and diethyl (methylthiomethyl)phosphonate (0.1 M) were prepared by diluting the reagents with milli-Q water (18 MΩ-cm). The concentrations were selected to be near the solubility limit or 1 M. Solutions of 2,2′ thiodiethanol and methylphosphonic acid were stored in the dark at room temperature and were replaced approximately every three months. Solutions of 2-chloroethyl ethyl sulfide, diethylcyanophosphonate, dimethylphosphite, and diethyl (methylthiomethyl)phosphonate were portioned into ~10 mL aliquots, stored at −60 °C to minimize decomposition [18], and replaced approximately every 3 months. Solutions were allowed to thaw at room temperature for 1 h before use.

Sensor (100 µL) was added to all the wells of a flat bottomed 96-well plate using a 12-channel electronic pipet. One hundred microliters of each analyte were applied to rows of the well plate. Thirty-two samples of water and 16 samples of the 7 analytes were tested to make the total sample size 144. The plate was scanned as detailed below.

All array images (24-bit color, 400 dpi) were collected using an Epson Perfection V700 desktop scanner in transparency mode. To eliminate interferences from stray light, the scanner was draped in black cloth. The images were analyzed with ImageJ [19], and the extraction of mean RGB values for each well was automated with a macro [20]. No attempts were made to correct for image-to-image variation by subtracting a control row, as one sensor was tested on each well plate, and our previous work found correcting images to be unnecessary [12].

All statistical analysis was performed using the statistical programming language R [21]. PCA was performed using the function prcomp. The data was mean-centered, but was not scaled to unit variance because all of the data were on a consistent scale of 0 to 255 RGB units. Score plots of the resulting data were constructed using ggbiplot library [22].

## 3. Results

### 3.1. Visible Colorimetric Changes

An actual image of the color changes of a warfare mimic is depicted in Figure 2, where an excerpt a 96-well plate of the colorimetric array is depicted. The first row shows the analyte 2-chloro ethyl ethylsulfide in the presence of 4 different sensors. Sensors A, B, C, D are Ellman’s reagent, fast blue B, alizarin yellow R, and eosin Y. In comparison to the control, all sensors have a different color in the presence of the analyte. 

### 3.2. Principle Component Analysis (PCA)

Principle component analysis (PCA) has been successfully used in the identification of explosives with colorimetric arrays [23]. Thus, images of the sensors using different analyte exposure were captured, and their red green blue (RGB) values were extracted for PCA analysis. This classification method has been established as a viable tool to interpret data generated by sensor arrays [11,14,24]. Figure 3 demonstrates the first two principle components of the RGB data that was obtained when all warfare mimics were reacted with the most effective 188 colorimetric sensors. It is seen that 2-chloro ethyl ethylsulfide, diethylcyanophosphonate, trifluoroacetic acid, methylphosphonic acid, and dimethylphosphite show clearly distinct clusters that are well separated from each other. Diethyl (methylthiomethyl)phosphonate and 2-2′-thiodiethanol are well clustered in each group, but not as well separated. However, PCA was still able to distinguish these two analytes from the control.

The PCA loading plot for the first component, shown in Figure 4, was used to determine which sensors and color channels contributed most to classifying the analytes that were screened with 188 sensors. The sensor channels that are centered around the 0.00 principal component loadings do not contribute effectively to classify the analytes, while the sensors that are located between +0.1 to +0.15 or −0.1 to 0.2 loadings, are effectively contributing to the analyte classification.

Table 1 identifies the sensors, the color channels of the sensors, and the PCA 1 and 2 loadings for each sensor. The PCA 1 and 2 loadings for each sensor indicated that alizarin yellow had 5 and the most loading contributions by the red and green color channels. This is followed by 3 contributions of the blue, green, and red color channels from the Ellman’s reagent, 3 contributions by the green and red channels from alizarin red, and 3 loadings by the red, green, and blue color channels from fast blue B. All other sensors only contributed one loading in one color channel.

For practical applications, the number of sensors should be minimized as much as possible, while still achieving appropriate selectivity. Four, eight, and twelve sensor arrays were simulated by making a subset of the RGB dataset, using sensors that had large contributions to the first and second components of the PCA analysis for the entire dataset. As shown in Figure 5, the twelve-sensor array shows well-separated groups, similar to the analysis of the entire dataset, except for an overlap of diethyl (methylthiomethyl)phosphonate and 2-2′-thiodiethanol with the water control. Similarly, the four- and eight-sensor arrays resulted in less separation between the organophosphorus mimics (data not shown). This suggests that at least twelve sensors would be needed in order to classify the warfare mimics most efficiently.

## 4. Discussions

Our results showed that the screening of 188 sensors for the detection of warfare mimics can generate RGB values that were analyzed by PCA to determine the top twenty sensors and color channels that were most effective in classifying 6 analytes that represented mustard gas and nerve agent warfare mimics. Furthermore, the entire dataset using 188 sensors was reduced to just twelve sensors showing very similar results as the large dataset. This is important because the miniaturization of colorimetric arrays allows for less sensor and analyte consumption, as well as quicker analysis of the RGB data. The first step of screening potential sensors for analytes like warfare mimics involves the qualitative analysis of sensor–analyte interactions. Therefore, only one analyte concentration between 0.1 M and 1.0 M was used in this study, which was previously described to give distinguishable results for various acids and bases [25]. A quantitative analysis of warfare analytes will be published soon, similar to what was described for printed sensors [26].

The color changes of the sensors in the array can been attributed to several mechanisms, including acid–base reactions, solvatochromic dye interactions, redox reactions, dipole–dipole interactions, and proton transfer reactions [27,28,29].

However, not all of the color changes are clearly understood. For example, one of the analytes, the mustard gas mimic, 2-chloro ethyl ethylsulfide, reacted with 5,5′-dithiobis-(2-nitrobenzoicacid), also known as the Ellman’s reagent (Figure 2). The Ellman’s reagent typically creates a yellow color change when a reaction occurs with thiols by the nucleophilic substitution on the disulfide bond [30]. However, in our study, a brown color change was induced by the mustard gas mimic, which is a sulfide, leading to a brown color change in comparison to the control (water). Figure 6 shows a potential mechanism that may still include a nucleophilic substitution of 2-chloro ethyl ethylsulfide (1) on the disulfide in the Ellman’s regent (2), potentially forming the brown colored cation (3), and the 2-carboxy-3-nitro-benzenethiol anion (4), in comparison to the control (water).

The future of these sensors includes the testing with real warfare agents and the deposition on paper-like substrates that is followed by imaging with smartphones. Colorimetric arrays that are on paper substrates may provide a low-cost alternative to other available sensors or lab-based instrumentation, making them a useful primary or substitute detection method when portable means are necessary. Thus, this technology has great potential for field deployment, where light, inexpensive, and easy-to-use technology is critical for identification of hazardous materials, like warfare agents. Additional applications for these versatile sensors include the analysis of street drugs by police officers or the detection of potentially dangerous compounds by first responders in emergency situations. The expedited process would provide these individuals with an effective method for testing and triaging threatening chemicals when their safety may rely on the speed of detection.

## Figures and Tables

**Figure 1 sensors-18-04291-f001:**
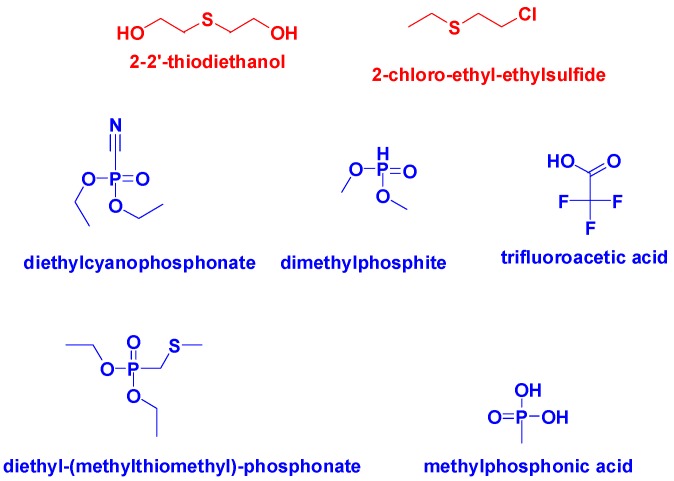
Chemical structures of mustard gas mimics (red) and nerve gas mimics (blue).

**Figure 2 sensors-18-04291-f002:**
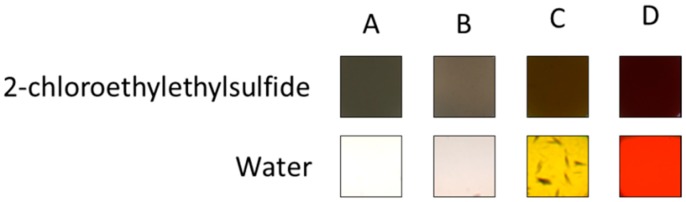
The top row shows the colorimetric changes of the mustard gas mimic 2-chloro ethyl ethylsulfide in the presence of Ellman’s reagent, fast blue B, alizarin yellow, and eosin Y (A, B, C, D). The bottom row shows the color changes in the presence of the control.

**Figure 3 sensors-18-04291-f003:**
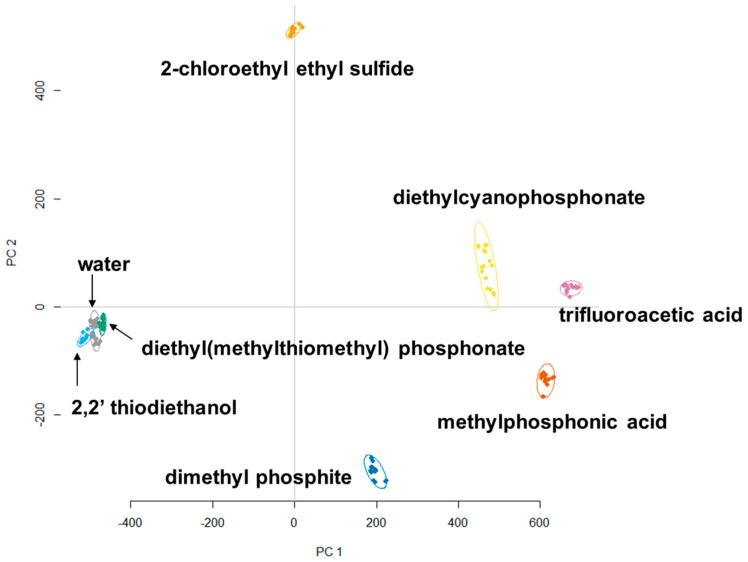
PCA plot showing the clustering of two mustard gas mimics and four nerve agent mimics. All analytes are clearly clustered into individual groups and distinguished from water.

**Figure 4 sensors-18-04291-f004:**
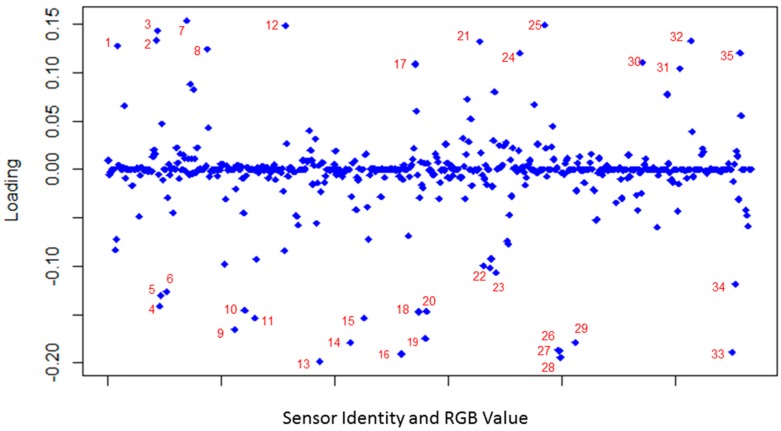
PCA loading plot demonstrating which sensors and color channels perform most effectively in analyte classification. The sensors and channels are (1) blue—2,4 dinitrophenol, (2) red—alizarin red S, (3) green—alizarin red S, (4) red—alizarin yellow R, (5) green—alizarin yellow R, (6) red—aurintricarboxylic acid, (7) red—bromocresol green, (8) red—chlorophenol red, (9) red—eosin Y, (10) red—erythrosin B, (11) red—fluorescein, (12) red—glycine cresol red, (13) red—methyl orange, (14) red—orange IV, (15) red—phloxine B, (16) red—thymol blue, (17) red—xylenol orange, (18) red—yellow A2, (19) red—alizarin yellow GG, (20) green—alizarin yellow GG, (21) red—iodophenol blue, (22) red—rosolic acid, (23) blue—4-(4-diethylaminophenylazo)pyridine, (24) blue—4-nitrocatechol, (25) red—nitrazine yellow, (26) red—Ellman’s reagent, (27) green—Ellman’s reagent, (28) blue—Ellman’s reagent, (29) red—metanil yellow, (30) blue—amanil fast yellow, (31) blue—thymolphthalein, (32) red—4-bromo-2,6-xylenol, (33) red—xylenol blue, (34) red—Rose bengal, (35) green—eosin B.

**Figure 5 sensors-18-04291-f005:**
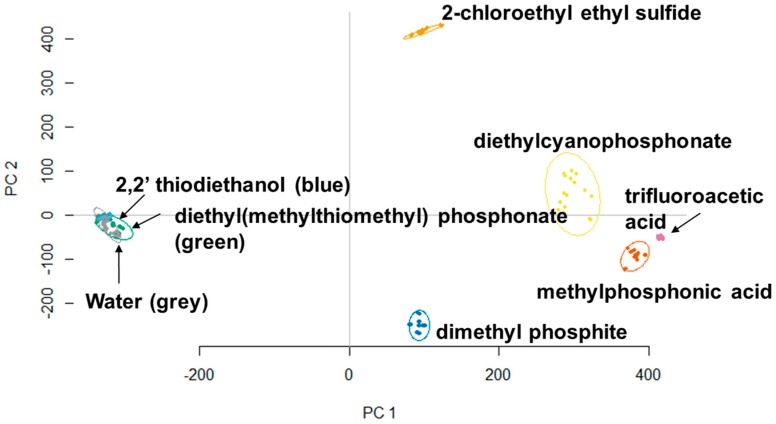
PCA plot of the reduced dataset using only the RGB data of twelve sensors. The sensors consisted of alizarin red S, alizarin yellow R, Ellman’s reagent, fast blue B, methyl orange, eosin Y, phloxine B, thymol blue, orange IV, morin, bromocresol purple, and xylenol blue.

**Figure 6 sensors-18-04291-f006:**
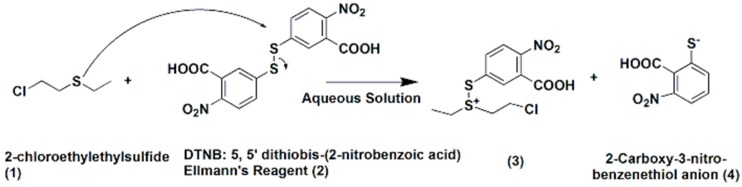
A possible mechanism of the reaction of mustard gas mimics 2-chloro ethyl ethylsulfide in the presence of Ellman’s reagent, leading to a dark brown color change. The nucleophilic substitution of the sulfur on the disulfide leads to proposed cationic product **1**, and the anionic leaving group forming a brown-colored product.

**Table 1 sensors-18-04291-t001:** The best performing sensors and color channels to classify two mustard gas mimics and four nerve agent mimics.

Sensor [Channel]	PC1 Loading	Sensor [Channel]	PC2 Loading
Methyl Orange [Red]	−0.198	Eosin Y [Red]	−0.274
Ellman’s Reagent [Blue]	−0.194	Phloxine B [Red]	−0.259
Thymol Blue [Red]	−0.191	Methyl Orange [Red]	0.223
Xylenol Blue [Red]	−0.188	Morin [Red]	−0.203
Ellman’s Reagent [Green]	−0.187	*o*-Dianisidine [Blue]	−0.202
Ellman’s Reagent [Blue]	−0.186	Bromocresol Purple [Red]	−0.201
Orange IV [Red]	−0.179	Bromothymol Blue [Red]	−0.199
Metanil Yellow [Red]	−0.179	Thymol Blue [Red]	−0.196
Alizarin Yellow GG [Red]	−0.174	*o*-Dianisidine [Green]	−0.195
Eosin Y [Red]	−0.165	*o*-Dianisidine [Red]	−0.194
Bromocresol Green [Red]	0.154	Morin [Green]	−0.187
Phloxine B [Red]	−0.153	Bromophenol blue [Red]	−0.180
Fluorescein [Red]	−0.153	Alizarin Red S [Green]	0.174
Nitrazine Yellow [Red]	0.149	Orange IV [Red]	0.170
Hexammine Cobalt (III) Chloride [Red]	0.148	Alizarin Red S [Red]	0.163
Yellow A2 [Red]	−0.147	Ferric Chloride [Blue]	−0.154
Alizarin Yellow GG[Green]	−0.146	Alizarin Yellow R [Red]	0.153
Erythrosin B [Red]	−0.145	1,1′-Diethyl-4,4′-Cyanine Iodide [Red]	−0.149
Alizarin Red S [Green]	0.143	Alizarin Yellow R [Green]	0.144
Alizarin Yellow R [Red]	−0.141	Chlorophenol Red [Red]	−0.135

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
