# Peer review of "The Identification of Seven Chemical Warfare Mimics Using a Colorimetric Array"

_sensors, 2018, doi:10.3390/s18124291_

Reviewer 1 Report

In this manuscript, the authors report a colorimetric array for the identification of chemical warfare agents. Due to the general applicability of this work, it is of interest to the sensor community. However, this work does not provide quantitative insight. I would therefore consider it favorable for publication in Sensors under the provision that the following concerns are adequately addressed.

[1] In your previous work (Anal. Chem. 2018, 90, 9990−9996), the colorimetric assay can identify and quantify analyte samples. However, in this work, it seems that there is only the qualitative analysis without the quantitative measurement. Is there a certain difficulty or challenge for the quantification of warfare agents? Please explain it.

[2] In Materials and Methods, I do not understand why the composition ratio (methyl red, methyl orange, phenolphthalein, and bromothymol blue) of the universal indicator is 1:3:7:8? Where does this ratio come from? experimental results or previous references?

[3] In Figure 5, it seems that PCA plot cannot discriminate two analytes (Diethyl-(methy-thiomethyl)-phosphonate and 2-2’-thiodiethanol) from the control. The authors should discuss this point and provide some methods to improve it.

Author Response

[1] In your previous work (Anal. Chem. 2018, 90, 9990−9996), the colorimetric assay can identify and quantify analyte samples. However, in this work, it seems that there is only the qualitative analysis without the quantitative measurement. Is there a certain difficulty or challenge for the quantification of warfare agents? Please explain it.

We added additional explanation and references to address this point:

The first step of screening potential sensors for analytes like warfare mimics involves the qualitative analysis of sensor-analyte interactions. Therefore, only one analyte concentration between 0.1 M and 1.0 M was used in this study, which was previously described to give distinguishable results for various acids and bases.[24] A quantitative analysis of warfare analytes will be published soon similar to what was described for printed sensors. [25]  

[2] In Materials and Methods, I do not understand why the composition ratio (methyl red, methyl orange, phenolphthalein, and bromothymol blue) of the universal indicator is 1:3:7:8? Where does this ratio come from? experimental results or previous references?

The text was modified and a reference was added.

The universal indicator was composed of Van Urk’s and Yamada’s recipe and contained methyl red, methyl orange, phenolphthalein, and bromothymol blue powders in a weight ratio of approximately 1:3:7:8.[17]

[3] In Figure 5, it seems that PCA plot cannot discriminate two analytes (Diethyl-(methy-thiomethyl)-phosphonate and 2-2’-thiodiethanol) from the control. The authors should discuss this point and provide some methods to improve it.

The text was modified as follows:

As shown in Figure 5, the twelve sensor array shows well separated groups similar to the analysis of the entire data set, except for an overlap of diethyl-(methy-thiomethyl)-phosphonate and 2-2’-thiodiethanol with the water control.Similarly, the four and eight sensor arrays resulted in less separation between the organophosphorus mimics (data not shown).  This suggests that at least twelve sensors would be needed in order to classify the warfare mimics most efficiently.   

Reviewer 2 Report

Following are my comments:

1) In general, the font size in all figures and tables are too small, which brings a lot of difficulty for reading

2) Is the Figure2 real image or just colored image? The author should clarify this point

3) For figure 4, what is the unit of loading? what is the label for x axis?

Author Response

1) In general, the font size in all figures and tables are too small, which brings a lot of difficulty for reading

All font sizes of the Figures and Table have been increased for easier readability.

2) Is the Figure2 real image or just colored image? The author should clarify this point

Figure 2 is a real image and additional text was added to make this clear.

"A actual image of the color changes of a warfare mimic is depicted in Figure 2".

3) For figure 4, what is the unit of loading? what is the label for x axis?

To our knowledge, there are no units for loading plots for the y axis. However, the x-axis is now labeled with the axis title: "Sensor Identity and RGB values:

Reviewer 3 Report

This manuscript describes a PCA approach to deduce a minimal array set of probes for identification of warfare using the mimics. The results are interesting, but the procedure is unclear. The followings are a list to be improved.

1. At 40th line, the full name for CDC

2. At 47th line, 188 probes were mentioned, but, in the SI, 189 names.

3. At 43rd line, 144 samples were mentioned and, at 79th line, 32 samples of water and 16 samples of the other analytes were told. The description about the samples were not clear and inconsistent, including the related results.

4. In the discussion, there are some comments on chemistry of the probes’ detection principle. However, it would be focused on the chemistry of 12 selected assays. Discussion of the chemical backgrounds for why these 12 reactions is enough for discrimination of the mimics would be something to be shown.

This is an interesting result, but to be corrected for better readability.  

Author Response

1. At 40th line, the full name for CDC

This has been changed to Center Disease Control

2. At 47th line, 188 probes were mentioned, but, in the SI, 189 names.

The last entry in the table (189) is not a sensor but the print stock. The 189 was removed and a note was made in the SI to make this clear.

3. At 43rd line, 144 samples were mentioned and, at 79th line, 32 samples of water and 16 samples of the other analytes were told. The description about the samples were not clear and inconsistent, including the related results.

There are 7 analytes  that were tested in 16 wells on two different well plates. That is 112 samples.  32 samples of water and 112 samples of analyte is 144 total samples.

Verbiage was added to clarify this point:

"32 samples of water and 16 samples of the 7 analytes were tested to make the total sample size 144"

4. In the discussion, there are some comments on chemistry of the probes’ detection principle. However, it would be focused on the chemistry of 12 selected assays. Discussion of the chemical backgrounds for why these 12 reactions is enough for discrimination of the mimics would be something to be shown.

The PCA analysis in this manuscript addresses what sensors work and not why the sensors work. A more extensive study would have to be done, such a NMR of sensor analyte complexes, or mass spectrometry of the products to address this point. In the future, these experiments will be done to learn more about which sensors react with what analytes and how. But at this point, the authors feel that this is beyond the scope of this paper.

Round  2

Reviewer 1 Report

I am satisfied with the revised version of the manuscript.

Reviewer 2 Report

publish

Reviewer 3 Report

I can accept the authors' response as the reasonable reply for my concerns.